# Topological magneto-optical effects and their quantization in noncoplanar antiferromagnets

Wanxiang Feng [1,2], Jan-Philipp Hanke [2,3], Xiaodong Zhou[1], Guang-Yu Guo[4,5], Stefan Blügel [2], Yuriy Mokrousov [2,3] & Yugui Yao[1]*

Reflecting the fundamental interactions of polarized light with magnetic matter, magneto-optical effects are well known since more than a century. The emergence of these phenomena is commonly attributed to the interplay between exchange splitting and spin-orbit coupling in the electronic structure of magnets. Using theoretical arguments, we demonstrate that topological magneto-optical effects can arise in noncoplanar antiferromagnets due to the finite scalar spin chirality, without any reference to exchange splitting or spin-orbit coupling. We propose spectral integrals of certain magneto-optical quantities that uncover the unique topological nature of the discovered effect. We also find that the Kerr and Faraday rotation angles can be quantized in insulating topological antiferromagnets in the low-frequency limit, owing to nontrivial global properties that manifest in quantum topological magneto-optical effects. Although the predicted topological and quantum topological magneto-optical effects are fundamentally distinct from conventional light-matter interactions, they can be measured by readily available experimental techniques.

[1] Key Lab of advanced optoelectronic quantum architecture and measurement (Ministry of Education), Beijing Key Lab of Nanophotonics & Ultrafine Optoelectronic Systems, and School of Physics, Beijing Institute of Technology, 100081 Beijing, China. [2] Peter Grünberg Institut and Institute for Advanced Simulation, Forschungszentrum Jülich and JARA, 52425 Jülich, Germany. [3] Institute of Physics, Johannes Gutenberg University Mainz, 55099 Mainz, Germany. [4] Department of Physics and Center for Theoretical Physics, National Taiwan University, Taipei 10617, Taiwan. [5] Physics Division, National Center for Theoretical Sciences, Hsinchu 30013, Taiwan. *email: ygyao@bit.edu.cn

Magneto-optical effects, referring to changes in the polarization state of light upon interacting with magnetic matter, are one of the most basic phenomena in solid-state physics. In 1846, Faraday discovered the first magneto-optical phenomenon for which the plane of linearly polarized light is rotated after passing through a piece of glass exposed to an external magnetic field[1]. Thirty years later, Kerr found the corresponding effect in the reflected light[2]. Magneto-optical effects, represented by the Faraday and Kerr effects, not only helped in establishing Maxwell's theory of electromagnetism in the late 19th century but provided also an exquisite technology for modern high-density data storage since the 1950s of last century[3,4]. Now, they have matured into appealing and widely used spectroscopic tools used to, e.g., visualize magnetic domains[5,6], detect and manipulate magnetic order[7,8], and measure ferromagnetism in two-dimensional (2D) systems[9,10].

The microscopic origin of the described magneto-optical effects has long been deemed to be the interplay between band exchange splitting (BES) and spin–orbit coupling (SOC)[11–18]. As an essential consequence of the Zeeman effect, BES is induced by either an external magnetic field or the spontaneous magnetization of magnetic materials. SOC further splits the bands so that the orbital motion of spin-polarized electrons couples to incident polarized light. The simultaneous presence of BES and SOC results in a different response to left-circularly and right-circularly polarized light in magnetic media as manifested in magneto-optical Faraday and Kerr effects. This microscopic mechanism has been the sole interpretation of magneto-optical effects until now.

Here, by using model arguments and first-principles calculations, we demonstrate that topological magneto-optical (TMO) effects, without any reference to BES or SOC, can arise in fully compensated noncoplanar antiferromagnets (nc-AFMs). The spectral integral of magneto-optical conductivity as well as the ones of Kerr and Faraday rotation angles, being spectroscopic fingerprints, identify the TMO effects essentially from their conventional cousins. Moreover, the quantum versions of these topological light–matter interactions, termed quantum topological magneto-optical (QTMO) effects, can be realized in insulating nc-AFMs with nontrivial topology in momentum space, for which the Kerr rotational angle is quantized to a certain value close to 90° and the Faraday rotation angle amounts to the product of Chern number and fine structure constant. The physical origin of TMO and QTMO effects is uncovered to be the nonzero scalar spin chirality (schematically shown in Fig. 1a), which differs fundamentally from any conventional light–matter coupling. The measurement of TMO and QTMO effects is feasible by the current experimental techniques.

## Result

**TMO effects.** Let us start with the TMO effects by considering the example of a three-dimensional (3D) face-centered-cubic (fcc) lattice, shown in Fig. 1b. We focus on the so-called $3Q$ spin structure[19], for which four magnetic sublattices form of a regular tetrahedron, and the spin on each sublattice points to the center of the tetrahedron, resulting in a fully compensated nc-AFM order. To describe conduction electrons interacting with the localized spin moments $\mathbf{S}_i$ ($|\mathbf{S}_i| = 1$), the Hamiltonian is expressed by the Kondo lattice model:

$$H = -\sum_{\langle ij \rangle} t_{ij} c_{i\alpha}^{\dagger} c_{j\alpha} - J \sum_i c_{i\alpha}^{\dagger} \boldsymbol{\sigma}_{\alpha\beta} c_{i\beta} \cdot \mathbf{S}_i. \tag{1}$$

Here, $c_{i\alpha}^{\dagger}$ ($c_{i\alpha}$) is the electron creation (annihilation) operator on site $i$ with spin $\alpha$, $\boldsymbol{\sigma}$ is the vector of Pauli matrices, $t_{ij}$ denotes the nearest-neighbor transfer integral, and $J$ is the on-site exchange coupling.

For the 3D fcc lattice, a proper strain can be used to generate a nonzero fictitious magnetic field[19], which effects the magneto-

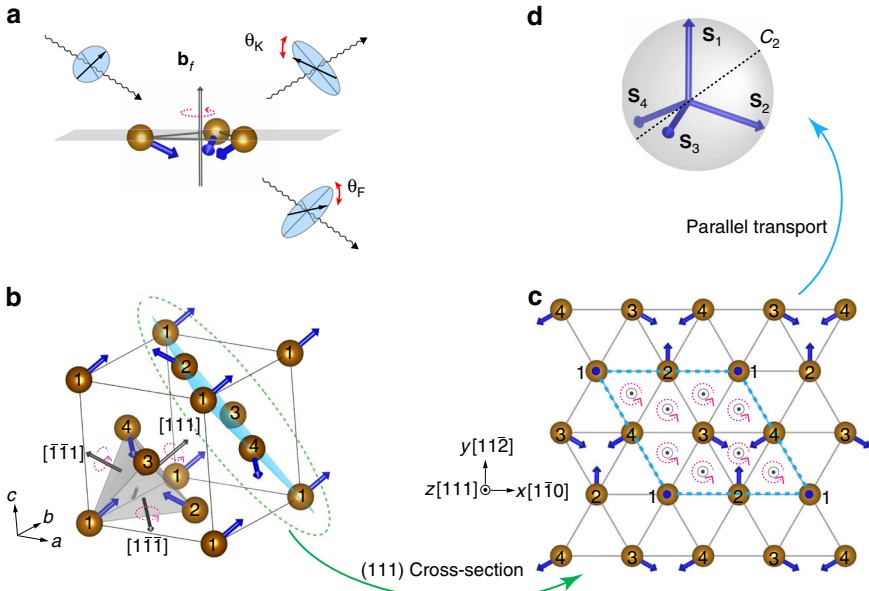

**Fig. 1 Topological light–matter interactions in chiral magnets. a** Sketch of topological light–matter interactions in a minimal chiral magnet comprising three neighboring noncoplanar spins. $\mathbf{b}_f$ is the fictitious magnetic flux generated by the finite scalar spin chirality. Topological magneto-optical Kerr and Faraday effects are induced by the fictitious magnetic flux $\mathbf{b}_f$ rather than by the nonzero net magnetization. **b** The $3Q$ noncoplanar antiferromagnet on a 3D fcc lattice. The numbers 1–4 label the four magnetic sublattices, which are the corners of a tetrahedron (gray color). Blue and gray arrows indicate the directions of the localized spin $\mathbf{S}_i$ and the fictitious magnetic flux $\mathbf{b}_f$, respectively. **c** The $3Q$ nc-AFM on a 2D triangular lattice. It is the (111) cross-section of the 3D fcc lattice, indicated by cyan color in **b**. Dashed cyan lines mark the 2D unit cell. ⊙ indicate the directions of the $\mathbf{b}_f$ on each triangular plaquette. **d** The unit sphere is spanned by the spins on four sublattices that are parallel transported to have a common origin. The dashed line indicates one of the $C_2$ rotation axes in spin space.

response properties of the system and ultimately leads to activating the TMO effect. Considering each face of the tetrahedron, the three noncoplanar spins provide a fictitious magnetic flux $\mathbf{b}_\mathrm{f} \propto t_3 \chi_{ijk} \hat{\mathbf{n}}_\mathrm{f}$, where $t_3 = t_{ij} t_{jk} t_{ki}$ is the successive transfer integral along a loop $i \to j \to k \to i$, $\chi_{ijk} = \mathbf{S}_i \cdot (\mathbf{S}_j \times \mathbf{S}_k)$ is the scalar spin chirality[19–21], and $\hat{\mathbf{n}}_\mathrm{f}$ is a unit vector normal to the face. This fictitious magnetic flux essentially comes from the orbital motion of electron because a Berry phase, being equivalent to the solid angle spanned by three neighboring spins, is picked up by the electron hopping along a closed loop on the triangular plaquette. The total fictitious magnetic field in the 3D fcc lattice is the vector sum of the magnetic fluxes on the four faces of the tetrahedron, i.e., $\mathbf{B} = \sum_{f=1}^{4} \mathbf{b}_\mathrm{f}$. In the unstrained case, $\mathbf{B}$ is zero because the four fluxes cancel each other exactly (Fig. 1b). After a uniaxial strain, characterized with a parameter $\delta$ (see the "Methods" section), is introduced along the [111] direction, the fictitious field is $\mathbf{B} = B \hat{\mathbf{n}}_{[111]}$ with $B \neq 0$ and the unit vector $\hat{\mathbf{n}}_{[111]}$ pointing into the [111] direction. The effect of this fictitious magnetic field is equivalent to the nonzero net magnetization in a ferromagnet or the external magnetic field applied to a non-magnet, and thus the response of a 3D fcc nc-AFM to the left-circularly and right-circularly polarized lights is inevitably different.

The role of strain can be understood in a more fundamental way from the symmetry point of view. In fact, the shape of linear response tensors can be fully determined by a group-theoretical analysis[22]. The magnetic point group of the 3D fcc nc-AFM is $m\bar{3}m'$[23], which suppresses the magneto-optical conductivity $\sigma_{xy}(\omega)$. The strain considered here removes all the symmetries containing fourfold rotations and the magnetic point group is lowered to $\bar{3}1m'$. As a consequence, this symmetry breaking facilitates nonvanishing magneto-optical conductivity, i.e., $\sigma_{xy}(\omega) = -\sigma_{yx}(\omega) \neq 0$. Note that the conductivity tensor $\sigma$ is a coordinate-dependent quantity. For the convenience, we choose the $[1\bar{1}0]$, $[11\bar{2}]$, and $[111]$ directions of the fcc lattice as the $x$, $y$, and $z$ axes, respectively (Fig. 1). Owing to the finite fictitious magnetic field that originates from the chiral spin structure in the strained case, the emergence of the transverse conductivity tensor components does not rely on the presence of SOC.

To confirm the symmetry analysis, we explicitly calculate the magneto-optical conductivity using the Kubo formula[24],

$$\sigma_{xy}(\omega) = \hbar e^2 \int \frac{d^3 k}{(2\pi)^3} \sum_{n \neq n'} (f_{n\mathbf{k}} - f_{n'\mathbf{k}})$$
$$\times \frac{\mathrm{Im}\left[ \langle \psi_{n\mathbf{k}} | v_x | \psi_{n'\mathbf{k}} \rangle \langle \psi_{n'\mathbf{k}} | v_y | \psi_{n\mathbf{k}} \rangle \right]}{(\epsilon_{n\mathbf{k}} - \epsilon_{n'\mathbf{k}})^2 - (\hbar\omega + i\eta)^2}, \tag{2}$$

where $v_i$ is the $i$th Cartesian component of the velocity operator, $\epsilon_{n\mathbf{k}}$ is the energy of the $n$th band at Bloch vector $\mathbf{k}$, $f_{n\mathbf{k}}$ is the Fermi–Dirac distribution function, $\hbar\omega$ is the photon energy, and $\eta$ is an adjustable smearing parameter. Figure 2b shows the real and imaginary parts of $\sigma_{xy}(\omega)$ computed for the model given by Eq. (1). It is evident that $\sigma_{xy}(\omega)$ turns out to be nonzero if the strain is applied ($\delta \neq 1$), and it increases with increasing $|\delta - 1|$. Upon inverting the direction of the applied strain, $\sigma_{xy}(\omega)$ changes its sign as the texture-induced emergent field is reversed. Since both $\sigma_{xy}^\mathrm{R}(\omega)$ and $B$ are proportional to $\delta$, as seen in Fig. 2c, the relation $\sigma_{xy}(\omega) \propto B$ can be verified. Based on nonzero $\sigma_{xy}(\omega)$, the coupling of this magnetic field to polarized light can manifest in Kerr and Faraday effects in chiral AFMs as described by Eqs. (8) and (9) below.

In the absence of SOC, the bands are spin degenerate regardless of the strain (Fig. 2a). This degeneracy is guaranteed by a fractional lattice translation combined with a pure spin rotation.

For example, the sublattices are exchanged ($1 \leftrightarrow 2, 3 \leftrightarrow 4$) under a fractional translation $(\mathbf{a}/2, \mathbf{b}/2, 0)$ along the [110] direction. After that, a spin rotation around the $C_2$ axis (Fig. 1d) restores the initial state ($\mathbf{S}_1 \leftrightarrow \mathbf{S}_2, \mathbf{S}_3 \leftrightarrow \mathbf{S}_4$). This degeneracy will be split by the SOC since in this case the spin is coupled to the lattice such that a pure spin rotation is not allowed.

The magneto-optical effects in 3D fcc nc-AFMs have a topological origin, in analogy to the topological Hall effect, since they are rooted in scalar spin chirality rather than SOC. More importantly, the BES is not a necessary condition for the emergence of magneto-optical effect, in contrast to the usual wisdom. The TMO effects we discovered here, requiring neither SOC nor BES, differ fundamentally from the conventional magneto-optical effects which have been intensively studied before. Therefore, the TMO effects have to be classified into the category of topological light–matter interactions.

**QTMO effects**. After uncovering the novel topological light–matter interactions in chiral AFMs, we now elucidate the intriguing cases in which the resulting TMO phenomena take quantized values. In particular, we will demonstrate that the characteristic Kerr and Faraday rotation angles amount to values that depend not on details of the electronic structure but rather on global nontrivial properties of the antiferromagnetic system. As an example, we consider a 2D nc-AFM with a triangular lattice and chiral 3Q spin structure as shown in Fig. 1c. By parallel transporting the four spins to have a common origin, we realize that they span a solid angle of $4\pi$ (Fig. 1d). It becomes intuitively clear that[23] the nontrivial spin texture can in principle give rise to quantized Hall transport[25]. The band structure and the anomalous Hall conductivity of the 2D triangular lattice are illustrated in Fig. 2d. As compared to the 3D fcc lattice, while the spin degeneracy of the bands remains, local band gaps occur between each pair of degenerate bands characterized by the Chern number $C = \pm 1$ and a quantized anomalous Hall conductivity $\sigma_{xy} = C e^2 / h$. The global band gap at 1/4 filling is closed when $J/t < 0.7$, while the one at 3/4 filling survives for arbitrarily small $J/t$ and hence holds a spontaneous quantum Hall effect[26,27]. This intriguing quantum state, originating from the spin chirality instead of the SOC, is called quantum topological Hall (QTH) insulator[28]. While the dc Hall conductivity due to spin chirality has been studied for the 2D triangular lattice before[26,27], here, we aim for its generalization in terms of a charge response to optical fields with finite frequency $\omega$, which has been overlooked so far. Since the magnetic point group of the 2D triangular lattice is $\bar{3}1m'$, nonzero $\sigma_{xy}(\omega)$ is indeed symmetry-allowed (Fig. 2e). This implies that the TMO effects mediated solely by the complex spin topology can exist in a QTH insulator.

Even more remarkably, the TMO effects emerging in the QTH insulators should be quantized in the low-frequency limit. The underlying physics is that for the QTH insulator, in analogy to the Chern insulator, the Maxwell's equations are modified by adding the magnetoelectric (axion) term $\frac{\Theta\alpha}{4\pi^2} \mathbf{E} \cdot \mathbf{B}$ into the usual Lagrangian[29]. The magnetoelectric polarizability (axion angle) $\Theta$ is quantized modulo $2\pi$ and in particular, $\Theta = \pi$ and $\Theta = 0$ classify topologically nontrivial and trivial insulators, respectively. By combining the modified Maxwell's equations and the free-standing slab geometry, one finds that in the low-frequency limit ($\hbar\omega \ll E_\mathrm{g}$, where $E_\mathrm{g}$ is the topologically nontrivial band gap) the Kerr and Faraday rotation angles in QTH insulators can be written as[30–32]

$$\theta_\mathrm{K} = -\tan^{-1}\left[ c / (2\pi \sigma_{xy}^\mathrm{R}) \right], \tag{3}$$

$$\theta_\mathrm{F} = \tan^{-1}(2\pi \sigma_{xy}^\mathrm{R} / c), \tag{4}$$

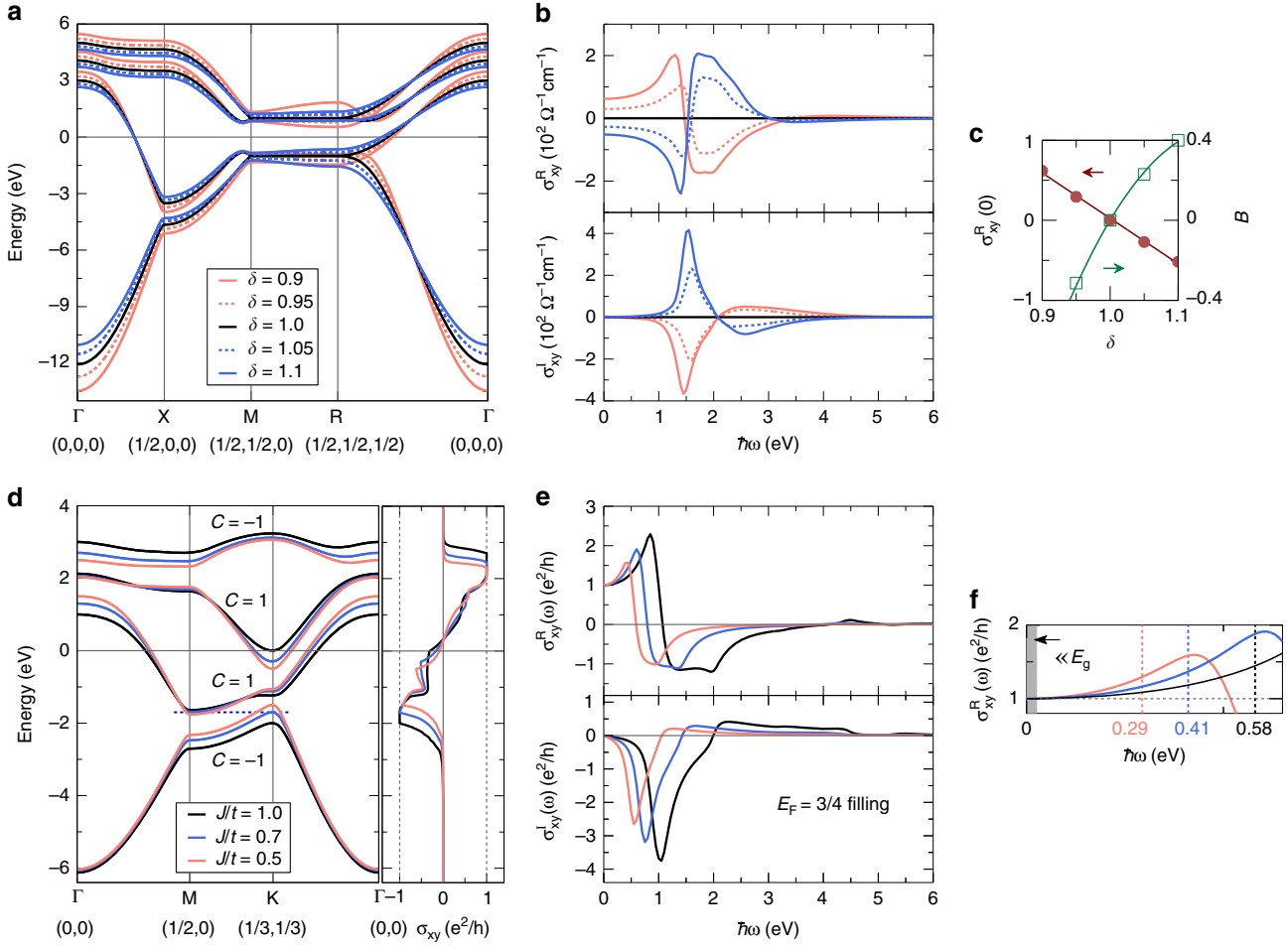

**Fig. 2 Electronic structure and magneto-optical conductivity of 3Q noncoplanar antiferromagnets. a** Band structure of the 3D fcc lattice ($J/t = 1.0$). **b** Magneto-optical conductivity of the 3D fcc lattice ($\eta = 0.1t$). Up and down panels show the real and imaginary parts with different $\delta$, respectively. **c** Both $\sigma_{xy}^{R}(\omega)$ (taking $\omega = 0$ as an example) and $B$ are proportional to $\delta$, verifying that $\sigma_{xy}(\omega) \propto B$. **d** Band structure and anomalous Hall conductivity of the 2D triangular lattice. **e** Magneto-optical conductivity of the 2D triangular lattice ($\eta = 0.1t$). The Fermi energy corresponds to the 3/4 band filling. **f** The enlarged low-frequency region of $\sigma_{xy}^{R}$. The dashed vertical lines mark the band gaps at 3/4 filling ($E_g$). The shaded area marks the low-frequency limit ($\hbar\omega \ll E_g$) in which the quantum topological magneto-optical effects arise.

where $c$ is the velocity of light in vacuum and $\sigma_{xy}^{R}$ is the real part of magneto-optical conductivity. Figure 2f clearly shows that $\sigma_{xy}^{R}(\omega \to 0) = Ce^2/h$ ($C = 1$ for the 2D triangular lattice)—thus, the quantized Kerr and Faraday angles occur inevitably with $\theta_K = -\tan^{-1}[1/(C\alpha)] \simeq -\pi/2$ and $\theta_F = \tan^{-1}(C\alpha) \simeq C\alpha$, where $\alpha = e^2/(\hbar c) \simeq 1/137$ is the fine structure constant. Recently, such kinds of quantized magneto-optical and magnetoelectric effects have been experimentally observed in various Chern insulator thin films[33–37].

**Realizing TMO and QTMO effects**. Armed with the above insights from the model analysis, we now consider real materials by taking $\gamma$-$Fe_xMn_{1-x}$ and $K_xRhO_2$ as prototypes that exhibit the TMO and the QTMO effects, respectively.

Disordered $\gamma$-$Fe_xMn_{1-x}$ alloys exhibit the multi-$Q$ spin texture in an fcc lattice, as evidenced by neutron diffraction measurements[38,39]. The 1$Q$ (Fig. 3a) and 2$Q$ (Fig. 3c) states are collinear AFMs which appear in the concentration range of $x < 0.4$ or $x > 0.8$, while the 3$Q$ state (Fig. 3b) as a noncollinear AFM exists when $0.4 < x < 0.8$. We first examine the electronic and magneto-optical properties in 3$Q$ spin texture (e.g., $\gamma$-$Fe_{0.5}Mn_{0.5}$). Under strain along the [111] direction, the band

structure without SOC remains doubly degenerate (see Supplementary Fig. 1), while the magneto-optical conductivity $\sigma_{xy}(\omega)$ turns out to be nonzero (see Supplementary Fig. 2). As a consequence, the Kerr and Faraday rotation angles, depicted in Fig. 3d, e, clearly depend on the strain. Their values are not changed after the SOC is switched on (see Supplementary Fig. 3, taking $\delta = 0.95$ as an example). This signifies the emergence of the TMO effects rooting entirely in the spin chirality, similarly to the topological orbital magnetization and topological Hall effect occurred in $\gamma$-$Fe_xMn_{1-x}$[40,41]. The discovery here differs from the case of the pyrochlore ferromagnet $Nd_2Mo_2O_7$[42], in which both the spin chirality and the SOC contribute to the Kerr effect. The magneto-optical strength (MOS) for the Kerr and Faraday effects, defined by $MOS_{K,F} = \int_0^{+\infty} \hbar |\theta_{K,F}(\omega)| d\omega$, can be used to analyze the whole trend of magneto-optical effects. Figure 3f illustrates the dependence of topological Kerr and Faraday effects on the strain. The linear relation can be established if the strain is sufficiently small.

TMO effects depend strongly on the spin texture and can be readily observed in experiments. The Kerr and Faraday angles are vanishing in 1$Q$ ($\theta = 0°$) and 2$Q$ ($\theta = 90°$) spin textures (Fig. 3g, h). If rotating the spins to form a nc-AFM ($0° < \theta < 90°$), the

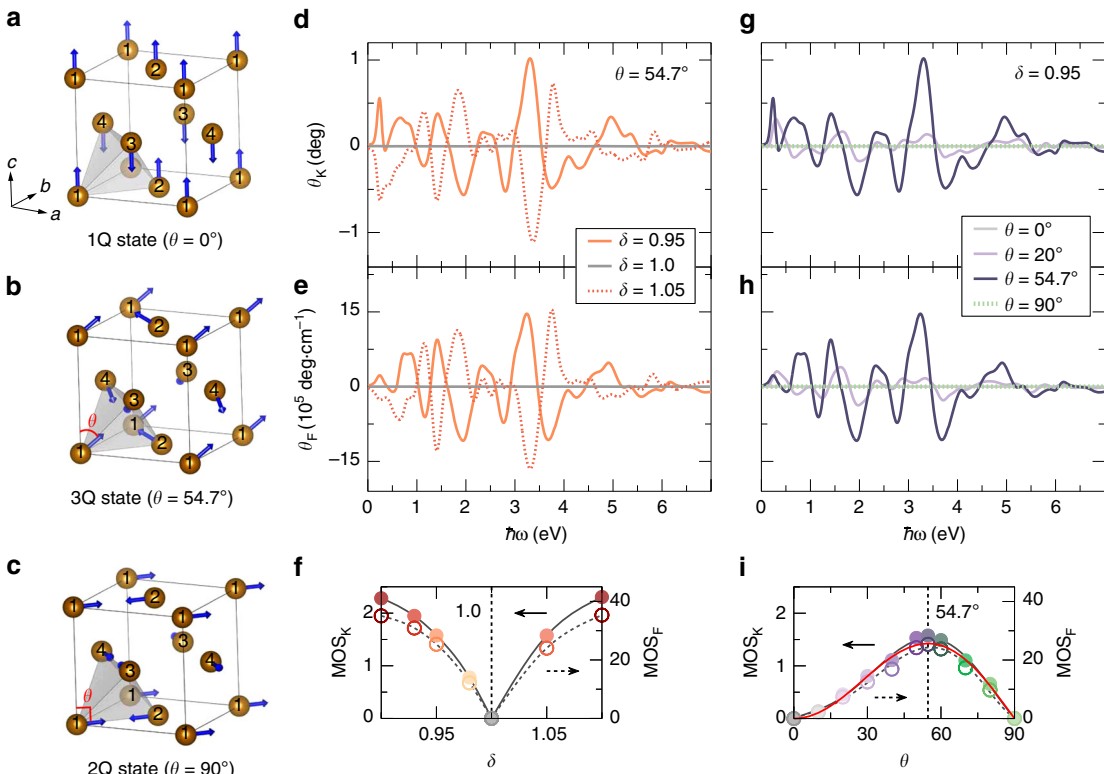

**Fig. 3 Topological magneto-optical effect in $\gamma$-Fe$_x$Mn$_{1-x}$. a–c** 1Q, 3Q, and 2Q spin textures of $\gamma$-Fe$_x$Mn$_{1-x}$. Blue arrows indicate the local spin moments. Numbers 1–4 label four magnetic sublattices (Fe or Mn site). The spin texture is characterized by the polar angle $\theta$ that measures the rotation of spins 1 and 2 within the $(1\bar{1}0)$ plane and the rotation of spins 3 and 4 within the (110) plane. **d**, **e** The Kerr and Faraday rotation angles of $\gamma$-Fe$_{0.5}$Mn$_{0.5}$ (3Q state) under various strain. **f** The magneto-optical strength (MOS) of Kerr and Faraday effects in $\gamma$-Fe$_{0.5}$Mn$_{0.5}$ (3Q state) as a function of strain. **g**, **h** The Kerr and Faraday rotation angles of $\gamma$-Fe$_x$Mn$_{1-x}$ with different spin textures. **i** The MOS of Kerr and Faraday effects in $\gamma$-Fe$_x$Mn$_{1-x}$ as a function of $\theta$. The solid red line denotes the angular dependence of scalar spin chirality, given by $\chi_{ijk}(\theta) \propto \cos\theta\sin^2\theta$. Spin–orbit coupling is not included in **d–i**.

nonzero spin chirality will arise, which in turn leads to nonzero Kerr and Faraday angles. Figure 3i shows that TMO effects are proportional to the scalar spin chirality and in particular, the Kerr and Faraday angles reach their maxima for 3Q spin texture ($\theta = 54.7°$). Since the $\gamma$-Fe$_x$Mn$_{1-x}$ thin film has been recently grown on the Cu/Al$_2$O$_3$ substrate along the (111) crystallographic direction and the lattice strain naturally occurs as compared to the bulk phase[41], we proclaim that TMO effects can be experimentally observed by varying the alloy concentration in $\gamma$-Fe$_x$Mn$_{1-x}$ thin film. The expected evidence is that the magneto-optical signals are suppressed in low and high concentrations, but will be activated for medium concentration since the spin chirality starts to play its role.

Next, we turn to another nc-AFM K$_x$RhO$_2$. It crystallizes in the $\gamma$-Na$_x$CoO$_2$-type structure where two RhO$_2$ monolayers (MLs) and two K ions layers stack alternately along the crystallographic $z$ axis[43–45]. The RhO$_2$ ML is a 2D nc-AFM in the sense that the Rh atoms are arranged on a triangular lattice with the 3Q spin structure (Fig. 4c). When $x = 0.5$, an insulating gap emerges because the $a_{1g}$ orbitals of the Rh$^{3.5+}$ (4$d^{5.5}$) ions have a filling factor of 3/4 under the trigonal deformation of RhO$_2$ octahedron. K$_{0.5}$RhO$_2$ was predicted to be a QTH insulator with the Chern number $C = 2$ because two RhO$_2$ MLs have identical spin structures (Fig. 4a, refer to the AA stacking) and each of them contributes $C = 1$[28]. When inverting all spins in one RhO$_2$ ML, the Chern number will flip its sign (i.e., $C = -1$). Hence, K$_{0.5}$RhO$_2$ having opposite spins in two RhO$_2$ MLs (Fig. 4b, refer to the AA' stacking) is a normal insulator with $C = 0$ (Supplementary Fig. 4).

In the absence of SOC, the bands are always doubly spin degenerate (Supplementary Figs. 4 and 5), however, the QTMO

effects can emerge in AA-stacked K$_{0.5}$RhO$_2$ and RhO$_2$ ML. Before assessing the magneto-optical effects, we need to calculate the optical conductivity. In Fig. 4e, $\sigma_{xy}^{R}$ displays a quantized behavior in the low-frequency limit, while $\sigma_{xx}^{R}$ (Fig. 4d) as well as the corresponding imaginary parts $\sigma_{xx}^{I}$ (Fig. 4f) and $\sigma_{xy}^{I}$ (Fig. 4g) tend to be zero when $\omega \to 0$. By plugging the optical conductivity into Eqs. (8), (9) and (12)–(15), we confirm the existence of QTMO effects, that is, $\theta_K \simeq -\pi/2$ and $\theta_F \simeq C\alpha$ in the low-frequency limit, as shown in Figs. 4h, i. To measure the quantized magneto-optical effects, the frequency of incident light should be much smaller than the topologically nontrivial band gap $E_g$[31,46]. The QTMO effects proposed in the RhO$_2$ ML could be experimentally observed if the frequency is below 3 THz ($\simeq 13$ meV). The powerful tool of time-domain THz spectroscopy is ready for exploring such kinds of quantized magneto-optical effects[33–37]. Thus, the quantized Kerr rotation angle should be measurable over a whole range of finite frequencies that extend up to the size of the nontrivial band gap (Fig. 4h).

**Spectroscopic fingerprints of TMO effects.** Unlike the topological Hall effect, the TMO effect accommodates additional information as it is a frequency-dependent quantity. Thus, the fundamental question arises whether there are characteristic features in the MO spectra that can distinguish the TMO effect from its trivial cousin. Inspired by the MOS (Fig. 3i), we discover that the integrals of $\sigma_{xy}^{R}(\omega)$, $\theta_K(\omega)$, and $\theta_F(\omega)$ are all proportional to the spin chirality $\chi_{ijk}(\theta)$. As a consequence, for the TMO effect, we propose the following three MO spectral integrals (SIs) to identify the signatures of the complex spin topology in the

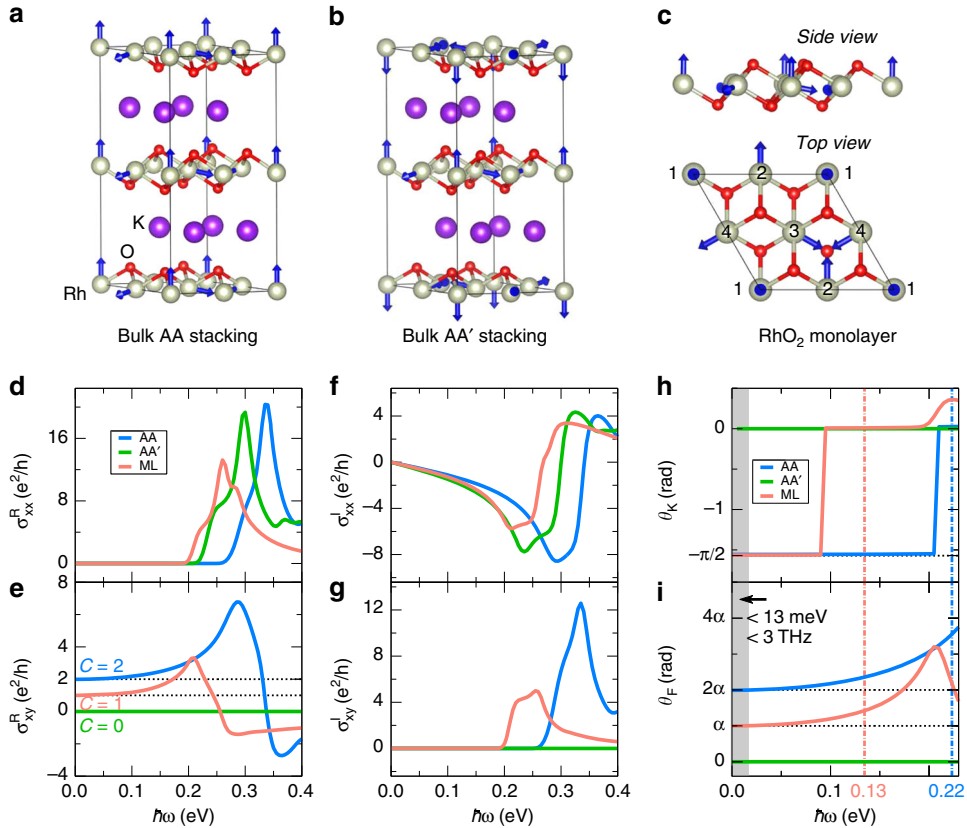

**Fig. 4 Quantum topological magneto-optical effect in $K_{0.5}RhO_2$. a, b** Bulk AA and AA′ stackings of $K_{0.5}RhO_2$. **c** Side and top views of $RhO_2$ monolayer (ML). **d–g** The optical conductivities of $K_{0.5}RhO_2$ and $RhO_2$ ML. The results of $K_{0.5}RhO_2$ are scaled by the length of crystallographic z axis. For $RhO_2$ ML, the Fermi energy is moved upward to the topologically nontrivial band gap $E_g$ (see Supplementary Fig. 5). **h, i** The Kerr and Faraday rotation angles of $K_{0.5}RhO_2$ and $RhO_2$ ML. The dash-dotted vertical lines indicate the $E_g$ of the AA-stacked $K_{0.5}RhO_2$ (0.22 eV) and $RhO_2$ ML (0.13 eV). The shaded area marks the low-frequency limit (e.g., $\hbar\omega < 0.1E_g^{ML} \simeq 13$ meV), in which the quantum topological magneto-optical effects can be measured by THz spectroscopy. Spin–orbit coupling is not included in **d–i**.

underlying spectra:

$$SI^{(1)} = \int_{0^+}^{\infty} \sigma_{xy}^R(\omega)d\omega \simeq K_R \chi_{ijk}(\theta), \qquad (5)$$

$$SI^{(2)} = \int_{0^+}^{\infty} \theta_K(\omega)d\omega \simeq K_K \chi_{ijk}(\theta), \qquad (6)$$

$$SI^{(3)} = \int_{0^+}^{\infty} \theta_F(\omega)d\omega \simeq K_F \chi_{ijk}(\theta), \qquad (7)$$

where $K_R$, $K_K$, and $K_F$ are scaling constants. Considering as an example the strained $\gamma$-$Fe_{0.5}Mn_{0.5}$ system, we demonstrate in Fig. 5a that the spectral integral $SI^{(1)}$ changes drastically with the magnetic order and the underlying spin topology. While this spectral integral follows the finite scalar spin chirality in the noncoplanar antiferromagnetic state, i.e., $SI^{(1)} \propto \chi_{ijk}(\theta) \propto \cos\theta\sin^2\theta$, its value relates instead to the magnetocrystalline anisotropy in the collinear ferromagnetic state. Specifically, the anisotropy function $K_0 + K_1\sin^2(\phi) + K_2\sin^4(\phi)$[47] with $\phi = \theta - 54.7°$ describes excellently the conventional MO spectrum of the hexagonal ferromagnet[48,49], which exhibits no spin chirality. As they directly relate to the MO conductivity, the SIs $SI^{(2)}$ and $SI^{(3)}$ for the Kerr and Faraday rotation angles reveal analogously in Fig. 5b, c fundamentally distinct behaviors for collinear and noncoplanar magnetic orders. Therefore, we proclaim that the different physical nature of the topological and conventional MO effects manifests in specific hallmarks in terms of the proposed SIs, which can thus be used to distinguish the two phenomena. Finally, while Fig. 5

promotes tuning the magnetic order to identify the spectroscopic fingerprints, we emphasize that applying strain is another suitable means. If we apply tensile or compressive strain, the TMO effect and the SIs change their signs in the noncoplanar antiferromagnet, which is in sharp contrast to the situation for the ferromagnetic state, where a sign change is purely accidental.

## Discussion

We discovered a fundamentally new type of light–matter interaction that originates from the chirality of the underlying complex spin texture of antiferromagnetic systems. As compared to the century-old microscopic interpretation based on the interplay of SOC and BES[11–18], the predicted TMO and QTMO effects mark a new class of solid-state phenomena that root in the concurrence of symmetry, chirality, and topology in magnetic materials. Thereby, we generalized the dc charge response[26,27] driven by noncoplanar magnetic order to the realm of nonzero frequency, which was unexplored so far. Specifically, we predicted the emergence of quantum topological Kerr effect with a quantized rotation angle of nearly 90°. The direct fingerprint of the complex spin texture of chiral magnets on the coupling to polarized light is thus significantly larger than in ferromagnetic materials, even though there might be no net spontaneous magnetization. Consequently, the proposed TMO and QTMO effects could be used to reveal domains of different chirality in general nc-AFMs. Although the topological light–matter interactions that we uncovered originate from fundamentally distinct physics, their manifestations in terms of changes of the

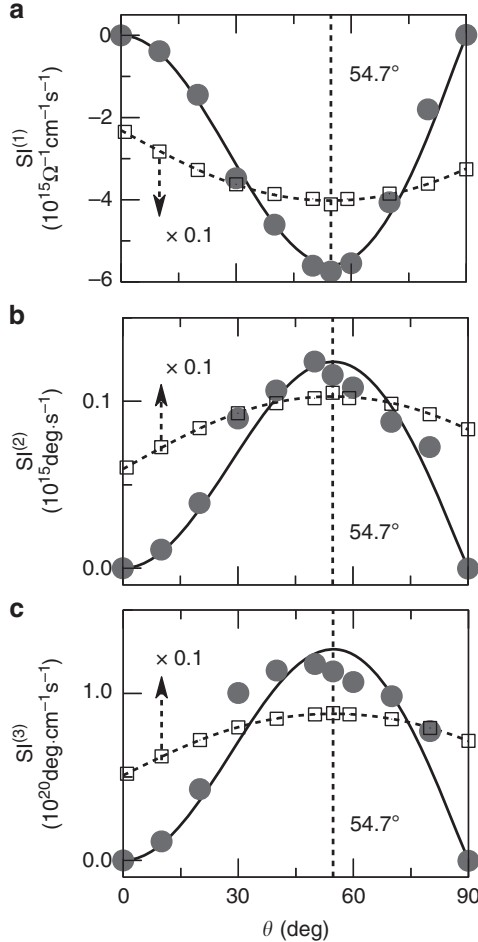

**Fig. 5 Spectroscopic hallmarks of the topological magneto-optical effect.** For the strained $\gamma$-Fe$_{0.5}$Mn$_{0.5}$ system with $\delta = 0.95$, the magnetic order as encoded in $\theta$ imprints on the spectral integrals of **a** the real part of the off-diagonal magneto-optical conductivity, and on the spectral integrals of **b** Kerr and **c** Faraday rotation angles. Solid circles and open squares represent the data for the chiral noncoplanar antiferromagnet and for the collinear ferromagnetic state, respectively. In the latter case, the magnetization changes with $\theta$ from the [001] to the [110] crystallographic direction. Spin–orbit coupling is included in the ferromagnetic case, for which the resulting integrals are divided by an overall factor of 10. The solid lines are fits of the obtained angular dependence to the scalar spin chirality, whereas the dashed lines are computed based on the magnetocrystalline anisotropy function with hexagonal symmetry, $K_0 + K_1 \sin^2(\phi) + K_2 \sin^4(\phi)$ where $\phi = \theta - 54.7°$, as the strain is applied along the [111] direction of cubic lattice.

polarization in reflected and transmitted light can be measured similarly to their conventional analogs by readily available experimental techniques.

While we assumed a polar geometry at normal incidence to predict novel types of chirality-driven magneto-optical effects, our conclusions are universal as they hold also for different measurement geometries. For example, in the case of $\gamma$-Fe$_x$Mn$_{1-x}$ the incident light propagates along the [111] direction parallel to the direction of the fictitious magnetic field due to finite scalar spin chirality. If the incident light deviates slightly from this direction, the Kerr angle acquires an additional geometrical factor of $\cos(\phi_i)/\cos(\phi_i \pm \phi_r)$[50] that depends on the light's polarization $\pm$, and includes the angles of incidence $\phi_i$ and reflectance $\phi_r$. Similarly, the Faraday angle at oblique incidence is approximately equal to the original one at normal incidence. The predicted

topological light–matter interactions are also active in both longitudinal and transversal geometries, for which the plane of incidence are the $zx$ and $xy$ planes (see Fig. 1), since they are also directly related to nonvanishing magneto-optical conductivity $\sigma_{xy}(\omega)$. In the case of the quantized TMO phenomena in K$_x$RhO$_2$, we studied explicitly the normal incidence of light that propagates along the [0001] direction. Since the long-wavelength and low-energy ($\lambda \gg l$ and $\hbar\omega \ll E_g$) predictions are independent of the angle of incidence[46], the quantization of the Kerr and Faraday rotation angles will be robust even under oblique incidence.

In this work, we explored the microscopic origin of TMO Kerr and Faraday phenomena, which are linear in the electric field, as representative light–matter interactions in chiral magnets. We anticipate that the scalar spin chirality imprints analogously on higher-order magneto-optical effects, including nonlinear Kerr (second-order in electric field) and Voigt phenomena (second-order in fictitious magnetic field **B**), and magnetic dichroism for both linearly and circularly X-ray polarized light (XMLD and XMCD). Our work advances the understanding and potential use of light–matter interactions in chiral magnets. Specifically, the discovered quantized versions of topological Kerr and Faraday effects are intimately linked to the quantized magneto-electric response of topological magnetic systems, realizing exotic axion electrodynamics[29,51]. Therefore, we promote the QTMO phenomena as an exciting platform to reveal and manipulate axion physics by coupling polarized light to the noncoplanar spin structure in antiferromagnetic materials. Ultimately, by exploring the coupling of polarized light to the spin pattern of antiferromagnetic materials, we also establish texture-driven magneto-optical effects as key physical phenomena in the emerging field of topological antiferromagnetic spintronics[52].

## Methods

**Expressions for TMO and QTMO effects.** The magneto-optical Kerr and Faraday effects measure the different response to left-circularly and right-circularly polarized light that propagates through a magnetic medium. Owing to this, the Kerr and Faraday angles are universally defined as:

$$\theta_K = \frac{1}{2}\left( \arg\{E_+^r\} - \arg\{E_-^r\} \right) \tag{8}$$

$$\theta_F = \frac{1}{2}\left( \arg\{E_+^t\} - \arg\{E_-^t\} \right) \tag{9}$$

where $E_\pm^{r,t} = E_x^{r,t} \pm iE_y^{r,t}$ are the left-circularly ($-$) and right-circularly ($+$) polarized components of the reflected (r) and transmitted (t) electric fields. Two distinct scenarios have to be considered separately. Case (1): In topologically trivial materials (e.g., $\gamma$-Fe$_x$Mn$_{1-x}$), by solving the conventional Maxwell's equations with appropriate boundary conditions, the complex Kerr angle in the polar geometry at normal incidence is given by[11–14]

$$\theta_K + i\epsilon_K = \frac{-\sigma_{xy}}{\sigma_{xx}\sqrt{1 + i(4\pi/\omega)\sigma_{xx}}}, \tag{10}$$

where $\theta_K$ and $\epsilon_K$ are the Kerr rotation angle and ellipticity, respectively. Similarly, the complex Faraday angle in the polar geometry at normal incidence reads[11–14]

$$\theta_F + i\epsilon_F = \frac{\omega l}{2c}(n_+ - n_-), \tag{11}$$

where $l$ is the thickness of the thin film, $c$ is the speed of light in vacuum, and $n_\pm = [1 + \frac{4\pi i}{\omega}(\sigma_{xx} \pm i\sigma_{xy})]^{1/2}$ are the complex refractive indices. Case (2): In the QTH insulators (e.g., K$_{0.5}$RhO$_2$), the Maxwell's equations have to be modified by the additional magnetoelectric term $\mathbf{E}\cdot\mathbf{B}$[29]. For a QTH insulator film with a thickness much shorter than the incoming light wavelength, the outgoing electric fields are derived as[31,46]

$$E_x^r = \left[1 - (1 + 4\pi\sigma_{xx})^2 - (4\pi\sigma_{xy})^2\right]A, \tag{12}$$

$$E_y^r = 8\pi\sigma_{xy}A, \tag{13}$$

$$E_x^t = 4(1 + 2\pi\sigma_{xx})A, \tag{14}$$

$$E_y^t = E_y^r, \tag{15}$$

with $A = 1 / \left[ (2 + 4\pi\sigma_{xx})^2 + (4\pi\sigma_{xy})^2 \right]$. Plugging Eqs. (12)–(15) into Eqs. (8) and (9), we obtain the quantized Kerr and Faraday angles in the low-frequency limit ($\omega \to 0$), which can be simply expressed by Eqs. (3) and (4).

**Tight-binding calculations**. The Kondo lattice model expressed in Eq. (1) is applicable for both the 3D fcc lattice and 2D triangular lattices. For the 3D fcc lattice, the strain is defined by $\delta = d'/d$, where $d'$ and $d$ refer to the distance between adjacent (111) planes in the strained and unstrained lattices, respectively. The transfer integral within (between) the (111) planes is given by $t_{ij} = t \, (t')$, where $t' = t/\delta^2$ is scaled with the strain ($t' = t$ implies no strain). An $8 \times 8$ matrix representation of the Hamiltonian was constructed by introducing eight orthonormal basis states $|i\alpha\rangle$ ($i = \{1, 2, 3, 4\}$, $\alpha = \{\uparrow, \downarrow\}$) that describes the interaction of itinerant electrons with the local spin moment $\mathbf{S}_i$. Using Fourier transformations, we transformed this matrix to a representation $H(\mathbf{k})$ in momentum space, which was subsequently diagonalized at every $\mathbf{k}$-point to access the band structure and magneto-optical conductivity.

**First-principles calculations**. The computational parameters of electronic structure: (1) $\gamma$-$Fe_xMn_{1-x}$. The self-consistent calculations were performed within the full-potential linearized augmented-plane-wave code FLEUR (see www.flapw.de). Exchange and correlation effects were treated in the generalized gradient approximation of the Perdew–Burke–Ernzerhof (GGA-PBE) functional[53]. The virtual crystal approximation was used to describe the disordered alloys by adapting the nuclear numbers under conservation of charge neutrality. The lattice constant of fcc $\gamma$-$Fe_xMn_{1-x}$ was chosen as 3.63 Å[40]. A compressive or tensile strain, $\delta = d'/d$, was applied along the [111] direction, where $d'$ and $d$ refer to the distance between adjacent (111) planes in the strained and unstrained lattices. The Poisson effect was accounted by the constant volume approximation. (2) $K_{0.5}RhO_2$. The self-consistent calculations were performed by the projector-augmented wave code VASP[54]. The GGA-PBE functional was used to treat exchange and correlation effects[53]. The GGA+U scheme with the effective Coulomb energy $U_{eff} = U - J = 2.0$ eV was applied for Rh $4d$ orbital to account for its Coulomb correlation effect[28]. The experimental lattice constants ($a = 3.065$ Å and $c = 13.600$ Å)[55] were used. A slab model with a vacuum region more than 15 Å was used for $RhO_2$ monolayer.

Using the WANNIER90 package[56], maximally localized Wannier functions were constructed based on the converged electronic structure in order to evaluate the optical conductivity tensor on an ultra-dense mesh of $k$-points. For metallic $\gamma$-$Fe_xMn_{1-x}$, the intraband contribution was considered by adding the phenomenological Drude term[12], $\sigma_D = \sigma_0 / (1 - i\omega\tau_D)$, into the diagonal element of optical conductivity. The Drude parameters ($\sigma_0$ and $\tau_D$) were obtained by linearly interpolating the experimental data of pure Fe ($\sigma_0 = 6.40 \times 10^{15}$ s$^{-1}$ and $\tau_D = 9.12 \times 10^{-15}$ s)[57] and pure Mn ($\sigma_0 = 4.00 \times 10^{15}$ s$^{-1}$ and $\tau_D = 0.33 \times 10^{-15}$ s)[58].

## Data availability

The tight-binding code and the data that support the findings of this study are available from the corresponding authors on reasonable request.

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

## Acknowledgements

The authors thank Di Xiao, Wang-Kong Tse, Branton J. Campbell, Hua Chen, and Fan Yang for fruitful discussions. W.F. and Y.Y. acknowledge the support from the National Key R&D Program of China (No. 2016YFA0300600) and the National Natural Science Foundation of China (Nos. 11734003, 11874085, and 11574029). W.F. also acknowledges the funding through an Alexander von Humboldt Fellowship. Y.M. and S.B. acknowledge funding under SPP 2137 "Skyrmionics" (project MO 1731/7-1), collaborative Research Center SFB 1238, and Y.M. acknowledges funding from project MO 1731/5-1 of Deutsche Forschungsgemeinschaft (DFG) and funding from the DFG via SFB/TRR 173. G.-Y.G. is supported by the Ministry of Science and Technology and the Academia Sinica as well as NCTS and Far Eastern Y.Z. Hsu Science and Technology Memorial Foundation in Taiwan. We acknowledge the Jülich Supercomputing Centre and RWTH Aachen University for providing computational resources under project jiff40.

## Author contributions

W.F. and Y.Y. conceived the research. W.F., X.Z. and J.-P.H. performed the first-principles calculations and model analysis. All authors contributed to the discussion of the data. W. F., J.-P.H., G.-Y.G. and Y.M. wrote the manuscript with discussion from S.B. and Y.Y.

## Competing interests

The authors declare no competing interests.
