## [Peer Review File · Nature Communications]

Reviewers' comments:

Reviewer #1 (Remarks to the Author):

In this paper, the authors showed that a finite scalar spin chirality can induce the topological magneto-optical effect even in the absence of the exchange splitting or spin-orbit coupling. They also showed that the Kerr and Faraday rotation angles can be quantized in the low-frequency limit. While the idea of topological (anti)ferromagnets is of great interest and is now a hot topic in condensed matter physics, the immediate relevance of this paper for the wide readership of Nature Communications is not clear and needs further justification. I believe that the importance of the present study should be justified more clearly.

(1) For the 3D fcc lattice, Shindou and Nagaosa showed that the anomalous Hall effect can arise when we introduce a distortion along the [111] or [110] direction (Ref. 19). Therefore, in Fig. 2b, it is not surprising that the optical conductivity is finite in the limit of zero frequency. Similarly, in Fig. 2d, it is not so surprising that the optical conductivity is finite at small frequencies.

On the other hand, in my opinion, what is new in this paper is the frequency dependence of the optical conductivity, which has not been investigated in previous studies on the anomalous Hall effect. The real part of the optical conductivity measures the average in the absorption of left- and right-circularly polarized light while the imaginary part measures the difference. It is an interesting question (for me) why the imaginary part has a sharp peak around 1.5eV in Fig.2(b) and 1 eV in Fig.2(d), and whether these peak structures fundamentally distinct from those due to conventional light-matter interactions. In other words, it would be very nice and extremely interesting if there is a frequency-dependence in the optical conductivity which characterize the topological magneto-optical effect, which cannot be observed in the usual magneto-optical effect. Similarly, I believe that the authors should discuss the frequency dependence of the optical conductivity in Figs.3, 4 and Figs. S2, S3.

(2) It is not surprising that the Kerr and Faraday rotation angles are quantized in the limit of zero frequency in quantum anomalous Hall insulators. While the authors claim that they discover quantum topological magneto-optical effect (QTMO), they always focus on the behaviors of these angles at low frequencies. Quantum anomalous Hall effect is interesting, but I believe that the authors should discuss the frequency dependence of the Kerr and Faraday rotation angles in more detail and clarify what is the characteristic feature of topological (anti)ferromagnets.

(3) Concerning the unit of the conductivity, $\text{Ohm}^{-1} \text{cm}^{-1}$ may be better, since the experimentalist usually use this unit.

I do not recommend this paper to be published in Nature Communications in the present form.

Reviewer #2 (Remarks to the Author):

The authors study magneto-optical effects in a model of spins coupled to conduction electrons and show that certain non-coplanar anti-ferromagnets produce novel Kerr and Faraday effects that depend on the scalar spin chirality of the spins. Moreover, these effects are quantized in dc limit to universal values. The authors further study two materials closely and show that the effects are measurable in them. Overall, I think the results are very interesting and elegant. In addition, the discussion is quite comprehensive.

My main complaint is regarding the novelty of the results. As the authors discuss themselves, similar

Hall conductivities, stemming from a flux induced by a scalar spin chirality, have been shown to occur on a triangular lattice in Ref 26 and 27. The current work describes 3D generalizations of the Refs 26, 27. The paper could be considered for publication if the authors reword the discussion taking this into account.

Reviewer #3 (Remarks to the Author):

The manuscript by W. Feng et.al. studied novel topological magneto-optical effects, Kerr and Faraday rotations by scalar spin chirality in non-coplanar antiferromagnets. They clearly demonstrated their idea in terms of the Kondo lattice model, where conduction electron spins and localized spins are coupled through the Kondo coupling. They showed that when the localized spins form a triple-Q structure on a triangle lattice with scalar spin chirality, the conduction electron band acquire non-trivial Berry curvature and non-zero Chern number. They demonstrated that due to the Berry curvature effect, the magnetic Kerr rotation angle becomes quantized to a certain value that is nearly the 90 degree. They further discuss their results in application to several real materials such as $\gamma\text{-Fe}_x\text{Mn}_{1-x}$ ($0.4 < x < 0.8$) and K_xRhO_2 ($x=0.5$). The topics is timely, the manuscript is clearly written, the relevance to the materials are quite promising, and scientific quantity is high enough for a publication in the Nature communications. Therefore, I strongly recommend the publication of the present manuscript in Nature communications. The following is a minor comment that the authors can take into account for their revision (optional).

In the text, I often encountered statements like 'Kerr rotational angle is quantized to the 90 degree exactly', e.g. the third sentence in the first paragraph of the discussion section. I am afraid that this expression is a little misleading to those general readers in the community of 'light-matter interaction' research. I suggest the authors to rewrite this into something like 'Kerr rotational angle is exactly quantized to a certain value, that is nearly the 90 degree.'

Response to Reviewer #1

Q: In this paper, the authors showed that a finite scalar spin chirality can induce the topological magneto-optical effect even in the absence of the exchange splitting or spin-orbit coupling. They also showed that the Kerr and Faraday rotation angles can be quantized in the low-frequency limit. While the idea of topological (anti)ferromagnets is of great interest and is now a hot topic in condensed matter physics, the immediate relevance of this paper for the wide readership of Nature Communications is not clear and needs further justification. I believe that the importance of the present study should be justified more clearly.

A: We thank the reviewer for his/her careful review of our manuscript. Magneto-optical (MO) effects are among the most basic physical phenomena in solid state physics. The simultaneous presence of band exchange-splitting (BES) and spin-orbit coupling (SOC) has been widely accepted to be the physical origin of the MO effects (e.g., Refs. 11-18 and references therein). The present work, for the first time, reveals that the topological magneto-optical (TMO) and quantum topological magneto-optical (QTMO) effects that rely neither on BES nor on SOC can emerge in noncoplanar antiferromagnets solely due to finite scalar spin chirality in real space. These exotic MO phenomena, decoding the microscopic interactions between light and chiral magnets, are fundamentally distinct from the conventional MO effects which have been intensively studied in collinear ferromagnets. Although we consider linear Kerr and Faraday effects as two prototypical examples to demonstrate the existence of TMO and QTMO effects, the topological light-matter interactions can be further extended to more general high-order MO effects. Our work deserves publication in Nature Communication since it has a wide readership as it marks a new class of solid-state phenomena that root in the concurrence of symmetry, chirality, and topology in magnetic materials.

Q: (1) For the 3D fcc lattice, Shindou and Nagaosa showed that the anomalous Hall effect can arise when we introduce a distortion along the [111] or [110] direction (Ref. 19). Therefore, in Fig. 2b, it is not surprising that the optical conductivity is finite in the limit of zero frequency. Similarly, in Fig. 2d, it is not so surprising that the optical conductivity is finite at small frequencies. On the other hand, in my opinion, what is new in this paper is the frequency dependence of the optical conductivity, which has not been investigated in previous studies on the anomalous Hall effect. The real part of the optical conductivity measures the average in the absorption of left- and right-circularly polarized light while the imaginary part measures the difference. It is an interesting question (for me) why the imaginary part has a sharp peak around 1.5eV in Fig.2(b) and 1 eV in Fig.2(d), and whether these peak structures fundamentally distinct from those due to conventional light-matter interactions. In other words, it would be very nice and extremely interesting if there is a frequency-dependence in the optical conductivity which characterize the topological magneto-optical effect, which cannot be observed in the usual magneto-optical effect. Similarly, I believe that the authors should discuss the frequency dependence of the optical conductivity in Figs.3, 4 and Figs. S2, S3.

A: Although the anomalous Hall (AH) effect may be interpreted as the zero-frequency limit of the MO effect, the both phenomena differ fundamentally from each other since they constitute two distinct families of physical effects that are driven either by electric fields or by optical fields. As an example, we note that the first observation of the AH effect in noncollinear antiferromagnet Mn_3Sn by Nakatsuji's group was published in Nature [527, 212 (2015)], and the first observation of the MO Kerr effect in Mn_3Sn also by Nakatsuji's group was subsequently published in Nature Photonics [12, 73 (2018)]. Both Nature and Nature Photonics are highly influential and competitive scientific journals. Our point is that the previous works on the AH effect in noncoplanar antiferromagnets do not affect the novelty of our present manuscript on the MO effect.

We thank the reviewer for his/her insightful suggestion of a more detailed analysis on the frequency dependence of the optical conductivity to identify fingerprints of the proposed topological MO effects. In response, we have carried out further numerical calculations and theoretical analyses in attempts to find out the characteristic features in the topological MO spectra that are distinctly different from that of conventional MO spectra. Below we present the main results of these further investigations.

(a) Tight-binding model. Since the tight-binding model we used here contains only eight bands that form spin-degenerate pairs, it is intuitively clear which states manifest in the absorption peaks. As noticed by the reviewer and shown in Fig. R1b (i.e., Fig. 2b in the main text), the MO spectrum exhibits an absorption peak at 1.5 eV. This feature originates from the interband transitions along the M-R path because there are two nearly flat bands with the energy separation of around 1.5 eV, as shown in Fig. R1a. We confirm this by explicit calculations of the momentum distribution of the “optical” Berry curvature on the $[\bar{1}\bar{1}0]$ plane, as plotted in Fig. R2a, which reveals hot spots close to the M-R line. In Fig. R1d (i.e., Fig. 2d in the main text), there is an absorption peak at 1.0 eV (for $J/t=1.0$). This peak correlates with interband transitions near the Γ point and along the Γ -M path, where the states from the third and the fourth degenerate bands are separated by about 1.0 eV (Note that we have put the Fermi level at $3/4$ band filling). Again, microscopic calculations of the “optical” Berry curvature shown in Fig. R2b verify that this absorption peak is dominated by the interband transitions on a hexagonal ring centered at the Γ point and also along the Γ -M paths.

To demonstrate that the discussed absorption peaks are clear signatures of the discovered TMO and QTMO effects due to the complex spin topology, we need to compare our results to the case of a collinear ferromagnet with zero scalar spin chirality. In this case, the presence of spin-orbit coupling (SOC) is crucial to observe any MO signal in the ferromagnetic state. However, the considered Kondo lattice model contains only one s orbital which exhibits no atomic SOC. While we investigated the possibility of other SOC terms generated by, for example, nearest-neighbor and next-nearest-neighbor hoppings, all of these vanish as the system is inversion symmetric with respect to the centers of the hopping lines (see the general definition of the possible SOC term in preprints arXiv:1802.03044, 1902.10650). As a consequence, the collinear model exhibits no MO effects such that the absorption peaks at 1.5 eV (Fig. 2b) and 1.0 eV (Fig. 2d) are a unique fingerprint of the non-trivial spin topology of the chiral antiferromagnet, as encoded in the proposed TMO and QTMO effects.

(b) First-principles calculations. Analogously to the model arguments, we consider the electronic structure of ferromagnetic states for the considered real materials. Owing to the SOC

for p and d orbitals, non-trivial MO spectra can appear in collinear ferromagnets.

In ferromagnetic γ -Fe_{0.5}Mn_{0.5}, all spins are set along the (111) direction (see Fig. R3b). Although the magnetic point groups are identical, the band structure of the ferromagnetic state changes a lot comparing with that of the 3Q noncoplanar antiferromagnetic state (c.f. Figs. R3c and R3d). Figs. R4a-d plots the optical conductivity of the collinear ferromagnetic and 3Q noncoplanar antiferromagnetic γ -Fe_{0.5}Mn_{0.5}. The diagonal elements (σ_{xx}^R and σ_{xx}^I) have a similar profile, while the positions and magnitudes of the peaks of the off-diagonal elements (σ_{xy}^R and σ_{xy}^I) are significantly different. For example, in the 3Q noncoplanar antiferromagnetic state, σ_{xy}^I amounts to the largest value of $-1 \times 10^3 \Omega^{-1} \text{cm}^{-1}$ at the photon energy of 1.45 eV. Seen from the “optical” Berry curvatures in Fig. R5a, we know that the interband transitions around the R point are responsible for this peak. On the other hand, in the collinear ferromagnetic state, the peak at the same photon energy is as small as $-0.22 \times 10^3 \Omega^{-1} \text{cm}^{-1}$ and the hot spots of “optical” Berry curvature move farther away from the R point (Fig. R5b). The interband transitions are greatly affected due to the existence of spin chirality, resulting to totally different MO spectra (Figs. R4e-f).

In the case of K_{0.5}RhO₂, the collinear ferromagnetic state is a metal (Figs. R6b and R6d), which differs from the insulating 3Q noncoplanar antiferromagnetic state (Figs. R6a and R6c). Moreover, the magnitudes of the diagonal and off-diagonal optical conductivities reduces significantly in the collinear ferromagnetic state (Fig. R7). Most importantly, σ_{xy}^R is not quantized in the limit of zero frequency as the electronic structure of the ferromagnetic state is no longer topologically nontrivial. In the 3Q noncoplanar antiferromagnetic state, the first peak of σ_{xy}^I ($\sim 3.6 e^2/h$) at the photon energy of 0.2 eV is contributed from the interband transitions around the K points (Fig. R8a). While the peak at the same photon energy in the collinear ferromagnetic state is one order of magnitude smaller ($\sim 0.3 e^2/h$), and thus the “optical” Berry curvature reduces accordingly (Fig. R8b).

In contrast to the tight-binding model, the electronic structure of the considered real materials is rather complex such that it is difficult to directly correlate the absorption peaks with the scalar spin chirality. Instead of analyzing the MO spectrum via its individual peaks, we thus exploit a powerful spectroscopic tool based on spectral integrals (SIs) that integrate the spectrum over frequency. Specifically, we use the following integrals of the real part of the off-diagonal MO conductivity as well as of the Kerr and Faraday rotation angles:

$$\text{SI}^{(1)} = \int_{0^+}^{\infty} \sigma_{xy}^R(\omega) d\omega \simeq K_R \chi_{ijk}(\theta),$$

$$\text{SI}^{(2)} = \int_{0^+}^{\infty} \theta_K(\omega) d\omega \simeq K_K \chi_{ijk}(\theta),$$

$$\text{SI}^{(3)} = \int_{0^+}^{\infty} \theta_F(\omega) d\omega \simeq K_F \chi_{ijk}(\theta),$$

where K_R , K_K , and K_F are scaling constants. Considering in Fig. R9 as an example the strained γ -Fe_{0.5}Mn_{0.5}, we find that these spectral properties can be used to tell apart the fundamentally distinct topological and conventional MO effects, which either stem from the complex spin topology of chiral antiferromagnets or relate to the magnetocrystalline anisotropy

of collinear ferromagnets.

In our revised manuscript, we use Fig. R1a as new Fig. 2a and we further add Fig. R9 as the additional Fig. 5 to the main text. Moreover, a new subsection ‘‘Spectroscopic fingerprints of TMO effects’’, which is our new discovery, has been also added to the revised manuscript (marked in blue color).

Fig. R1 (a) Band structure of the 3D fcc lattice ($J/t = 1.0$). It differs from original Fig. 2a in the main text by the different choice of k -paths. (b) Magneto-optical conductivity of the 3D fcc lattice ($\eta = 0.1t$). (c) Band structure and anomalous Hall conductivity of the 2D triangular lattice. (d) Magneto-optical conductivity of the 2D triangular lattice ($\eta = 0.1t$).

Fig. R2 (a) Optical Berry curvature of the 3D fcc lattice at the $(1\bar{1}0)$ plane by fixing the photon energy at 1.5 eV. (b) Optical Berry curvature of the 2D triangular lattice by fixing the photon energy at 1.0 eV. The unit of Berry curvature is Bohrs^2 .

Fig. R3. (a,c) Crystal and band structures of 3Q noncoplanar antiferromagnetic $\gamma\text{-Fe}_{0.5}\text{Mn}_{0.5}$. (b,d) Crystal and band structures of collinear ferromagnetic $\gamma\text{-Fe}_{0.5}\text{Mn}_{0.5}$. SOC is included for band structure calculations.

Fig. R4. (a-d) Optical conductivity of collinear ferromagnetic (FM) and 3Q noncoplanar antiferromagnetic γ - $\text{Fe}_{0.5}\text{Mn}_{0.5}$. (e,f) Kerr and Faraday rotation angles of collinear ferromagnetic and 3Q noncoplanar antiferromagnetic γ - $\text{Fe}_{0.5}\text{Mn}_{0.5}$. The results are included SOC. *The data of 3Q noncoplanar antiferromagnetic γ - $\text{Fe}_{0.5}\text{Mn}_{0.5}$ are taken from Figs. 3, S2, and S3.*

Fig. R5. Optical Berry curvatures of 3Q noncoplanar antiferromagnetic (a) and collinear ferromagnetic (b) γ - $\text{Fe}_{0.5}\text{Mn}_{0.5}$ at the $(1\bar{1}0)$ plane by fixing the photon energy at 1.45 eV.

Fig. R6. (a,c) Crystal and band structures of 3Q noncoplanar antiferromagnetic $K_{0.5}RhO_2$. (b,d) Crystal and band structures of collinear ferromagnetic $K_{0.5}RhO_2$. SOC is included for band structure calculations.

Fig. R7. (a-d) Optical conductivity of collinear ferromagnetic (FM) and 3Q noncoplanar antiferromagnetic $K_{0.5}RhO_2$. (e,f) Kerr and Faraday rotation angles of collinear ferromagnetic and 3Q noncoplanar $K_{0.5}RhO_2$.

antiferromagnetic $K_{0.5}RhO_2$. α in (f) is fine structure constant. The results are included SOC.

Fig. R8. Optical Berry curvatures of 3Q noncoplanar antiferromagnetic (a) and collinear ferromagnetic (b) $K_{0.5}RhO_2$ at the $k_z=0$ plane by fixing the photon energy at 0.2 eV.

Fig. R9. Spectroscopic hallmarks of the topological magneto-optical effect. For the strained $\gamma\text{-Fe}_{0.5}\text{Mn}_{0.5}$ system with $\delta = 0.95$, the magnetic order as encoded in θ imprints on the spectral integrals of (a) the real part of the off-diagonal magneto-optical conductivity, and on the spectral integrals of (b) Kerr and (c) Faraday rotation angles. Solid circles and open squares represent the data for the chiral noncoplanar antiferromagnet and for the collinear ferromagnetic state, respectively. In the latter case, the magnetization

changes with θ from the [001] to the [110] crystallographic direction. Spin-orbit coupling is included in the ferromagnetic case, for which the resulting integrals are divided by an overall factor of 10. The solid lines are fits of the obtained angular dependence to the scalar spin chirality, whereas the dashed lines are computed based on the magnetocrystalline anisotropy function with a hexagonal symmetry, $K_0 + K_1 \sin^2 \phi + K_2 \sin^4 \phi$ ($\phi = \theta - 54.7^\circ$), as the strain is applied along the [111] direction of cubic lattice.

Q: (2) It is not surprising that the Kerr and Faraday rotation angles are quantized in the limit of zero frequency in quantum anomalous Hall insulators. While the authors claim that they discover quantum topological magneto-optical effect (QTMO), they always focus on the behaviors of these angles at low frequencies. Quantum anomalous Hall effect is interesting, but I believe that the authors should discuss the frequency dependence of the Kerr and Faraday rotation angles in more detail and clarify what is the characteristic feature of topological (anti)ferromagnets.

A: Comment (2) appears to be similar (or related) to comment (1). Therefore, in addition to our response to comment (1) above, we would like to add the following explanations. We agree with reviewer #1 that the Kerr and Faraday rotation angles are quantized in the low frequency region *but not only the limit of zero frequency*. In particular, Fig. 4h shows that the Kerr rotation angle of $K_{0.5}\text{RhO}_2$ is quantized up to the photon energy of ~ 0.20 eV, being close to its nontrivial band gap, and thus the QTMO effect can be measured by using the MO Kerr rotation spectroscopy at a finite (“not too low”) frequency.

Furthermore, we would emphasize that although the quantized Kerr and Faraday rotation angles in quantum anomalous Hall insulators can be called quantum magneto-optical (QMO) effect, the QTMO effect we report in the present manuscript is fundamentally different. This previously unexplored effect differs from the QMO effect in that the latter arises from the SOC in a ferromagnet while the former is due to scalar spin chirality. To further illustrate our point, we list the families of the AH and the MO effects in Table R1, from which one can see that the QTMO and QMO effects do not belong to the same family. This point has been further strengthened by the fact that their spectral integrals are fundamentally distinct by respectively relating to scalar spin chirality and magnetocrystalline anisotropy, as we presented above.

Table R1. The family members of the anomalous Hall (AH) and magneto-optical (MO) effects as well as their distinctly different origins. QAH: quantum anomalous Hall; QMO: quantum magneto-optical; TH: topological Hall; TMO: topological magnet-optical; QTH: quantum topological Hall; QTMO: quantum topological magneto-optical.

	AH family (driven by an electric field)	MO family (driven by an optical field)
Spin-orbit coupling + band spin splitting	AH effect ^a	MO effect ^e
	QAH effect ^b	QMO effect ^f
Scalar spin chirality	TH effect ^c	TMO effect [*]
	QTH effect ^d	QTMO effect [*]

^aNagaosa *et al.*, Rev. Mod. Phys. 82, 1539 (2010) and references therein;

^bLiu *et al.*, Annu. Rev. Condens. Matter Phys. 7, 301 (2016) and references therein;

^cRefs. 19-20 and references therein; ^dRef. 28; ^eRefs. 11-14 and references therein;

^fRefs. 30-32 (theory) and Refs. 33-37 (experiment);

^{*}The present work.

Q: (3) Concerning the unit of the conductivity, $\text{Ohm}^{-1} \text{cm}^{-1}$ may be better, since the experimentalist usually use this unit.

A: We thank the reviewer for this helpful remark. While we follow the reviewer's suggestion to convert the unit of optical conductivity to $\Omega^{-1}\text{cm}^{-1}$ in the 3D systems, we keep the unit Ω^{-1} (i.e., e^2/h) for the optical conductivity in the 2D systems. In analogy to charge transport due to electric fields in 2D, the unit chosen in the latter case is more natural to represent the potentially quantized character of transport due to optical fields.

Q: I do not recommend this paper to be published in Nature Communications in the present form.

A: We again thank the reviewer for his/her valuable comments and helpful suggestions, which we have considered carefully. We hope that our revised manuscript (marked in **blue** color) together with the detailed explanations above, will be judged by the reviewer to be suitable for publication in Nature Communications.

 Response to Reviewer #2

Q: The authors study magneto-optical effects in a model of spins coupled to conduction electrons and show that certain non-coplanar anti-ferromagnets produce novel Kerr and Faraday effects that depend on the scalar spin chirality of the spins. Moreover, these effects are quantized in dc limit to universal values. The authors further study two materials closely and show that the effects are measurable in them. Overall, I think the results are very interesting and elegant. In addition, the discussion is quite comprehensive.

A: We thank the reviewer for his/her careful review of our manuscript, and appreciate the reviewer’s assessment that our results on “novel Kerr and Faraday effects” are overall “very interesting and elegant”. We hope that we addressed adequately the reviewer’s remark below such that the reviewer can recommend our revised manuscript for publication.

Q: My main complaint is regarding the novelty of the results. As the authors discuss themselves, similar Hall conductivities, stemming from a flux induced by a scalar spin chirality, have been shown to occur on a triangular lattice in Ref 26 and 27. The current work describes 3D generalizations of the Refs 26, 27. The paper could be considered for publication if the authors reword the discussion taking this into account.

A: We thank the reviewer for this insightful comment but we cannot share the formulated concern about the novelty of our results. While we indeed mention the topological Hall effect induced by the scalar spin chirality on 2D and 3D lattices (see, e.g., Refs. 26 and 27), this discussion serves only as an intuitive starting point for the present work, preparing the ground for our main findings, namely, the emergence of TMO and QTMO effects due to complex spin topology. As mentioned by the reviewer, our work uncovers novel Kerr and Faraday effects. Thereby, we add new members to the family of response phenomena, which are fundamentally distinct from the topological Hall effects as illustrated in Tab. R1. Nevertheless, considering the reviewer’s suggestion, we reformulated the discussion on pages 4 and 7 in the revised manuscript (see changes of main text marked in **blue** color) to underline that our present work can be considered as the generalization of the works by Kato, Martin, and Batista (Refs. 26 and 27).

Table R1. The family members of the anomalous Hall (AH) and magneto-optical (MO) effects as well as their distinctly different origins. QAH: quantum anomalous Hall; QMO: quantum magneto-optical; TH: topological Hall; TMO: topological magnet-optical; QTH: quantum topological Hall; QTMO: quantum topological magneto-optical.

	AH family (driven by an electric field)	MO family (driven by an optical field)
Spin-orbit coupling + band spin splitting	AH effect ^a	MO effect ^e
	QAH effect ^b	QMO effect ^f
Scalar spin chirality	TH effect ^c	TMO effect [*]
	QTH effect ^d	QTMO effect [*]

^aNagaosa *et al.*, Rev. Mod. Phys. 82, 1539 (2010) and references therein;

^bLiu *et al.*, Annu. Rev. Condens. Matter Phys. 7, 301 (2016) and references therein;

^cRefs. 19-20 and references therein; ^dRef. 28; ^eRefs. 11-14 and references therein;

^fRefs. 30-32 (theory) and Refs. 33-37 (experiment);

*The present work.

Response to Reviewer #3

Q: The manuscript by W. Feng et.al. studied novel topological magneto-optical effects, Kerr and Faraday rotations by scalar spin chirality in non-coplanar antiferromagnets. They clearly demonstrated their idea in terms of the Kondo lattice model, where conduction electron spins and localized spins are coupled through the Kondo coupling. They showed that when the localized spins form a triple-Q structure on a triangle lattice with scalar spin chirality, the conduction electron band acquire non-trivial Berry curvature and non-zero Chern number. They demonstrated that due to the Berry curvature effect, the magnetic Kerr rotation angle becomes quantized to a certain value that is nearly the 90 degree. They further discuss their results in application to several real materials such as $\gamma\text{-Fe}_x\text{Mn}_{1-x}$ ($0.4 < x < 0.8$) and K_xRhO_2 ($x=0.5$). The topics is timely, the manuscript is clearly written, the relevance to the materials are quite promising, and scientific quantity is high enough for a publication in the Nature communications. Therefore, I strongly recommend the publication of the present manuscript in Nature communications. The following is a minor comment that the authors can take into account for their revision (optional).

A: We thank the reviewer for his/her valuable time, the excellent summary of our work, and the strong recommendation for publication of our manuscript in Nature Communications. In particular, we appreciate the reviewer's assessment that our study of "novel topological magneto-optical effects" is "timely" and that the considered materials are "quite promising".

Q: In the text, I often encountered statements like 'Kerr rotational angle is quantized to the 90 degree exactly', e.g. the third sentence in the first paragraph of the discussion section. I am afraid that this expression is a little misleading to those general readers in the community of 'light-matter interaction' research. I suggest the authors to rewrite this into something like 'Kerr rotational angle is exactly quantized to a certain value, that is nearly the 90 degree.'

*A: We thank the reviewer for this helpful suggestion, which we followed in our revised manuscript (see changes of main text marked in **blue** color).*

REVIEWERS' COMMENTS:

Reviewer #1 (Remarks to the Author):

I agree that the authors responded properly to all the criticisms and comments raised in the previous report. I would like to recommend this paper to be published in Nature Communications.

Reviewer #2 (Remarks to the Author):

The authors have satisfactorily addressed my concerns. I now recommend it for publication in Nature Comm.

Reviewer #3 (Remarks to the Author):

I have read the reply by the authors. The author replied my comments properly. I am happy to recommend a publication of the current manuscript in your journal.